# A New Look on Long-COVID Effects: The Functional Brain Fog Syndrome

**DOI:** 10.3390/jcm11195529

**Published:** 2022-09-21

**Authors:** Maria Donata Orfei, Desirée Estela Porcari, Sonia D’Arcangelo, Francesca Maggi, Dario Russignaga, Emiliano Ricciardi

**Affiliations:** 1Molecular Mind Laboratory (MoMiLab), IMT School for Advanced Studies Lucca, Piazza S. Francesco 19, 55100 Lucca, Italy; 2Intesa Sanpaolo Innovation Center SpA Neuroscience Lab, Via Inghilterra 3, 10138 Turin, Italy; 3Intesa Sanpaolo S.p.A., HSE Office, Via Lorenteggio 266, 20152 Milan, Italy

**Keywords:** COVID-19, pandemic, brain fog, subjective cognitive complaints, long haulers

## Abstract

Epidemiological data and etiopathogenesis of brain fog are very heterogeneous in the literature, preventing adequate diagnosis and treatment. Our study aimed to explore the relationship between brain fog, neuropsychiatric and cognitive symptoms in the general population. A sample of 441 subjects underwent a web-based survey, including the PANAS, the DASS-21, the IES-R, the Beck Cognitive Insight Scale, and a questionnaire investigating demographic information, brain fog, subjective cognitive impairments (Scc) and sleep disorders. ANOVA, ANCOVA, correlation and multiple stepwise regression analyses were performed. In our sample, 33% of participants were defined as Healthy Subjects (HS; no brain fog, no Scc), 27% as Probable Brain Fog (PBF; brain fog or Scc), and 40% as Functional Brain Fog (FBF; brain fog plus Scc). PBF and FBF showed higher levels of neuropsychiatric symptoms than HS, and FBF showed the worst psychological outcome. Moreover, worse cognitive symptoms were related to the female gender, greater neuropsychiatric symptoms, sleep disorders, and rumination/indecision. Being a woman and more severe neuropsychiatric symptoms were predictors of FBF severity. Our data pointed out a high prevalence and various levels of severity and impairments of brain fog, suggesting a classificatory proposal and a multifaceted etiopathogenic model, thus facilitating adequate diagnostic and therapeutic approaches.

## 1. Introduction

Long-COVID syndrome consists of many persistent symptoms following COVID-19 illness, which can endure for several weeks or months [1]. It is estimated to affect at least one-third of patients, with a prevalence increasing in patients hospitalized because of COVID-19 infection [2,3]. One of the most frequent Long-COVID effects is brain fog [4], a colloquial expression indicating a phenomenon whose clinical impact may be detrimental to an individual’s psychological, occupational, and social life. Most subjects affected by brain fog report forgetfulness, feeling confused, having difficulties concentrating, decision-making, working, and performing daily activities, not to say social relationships and communication [5]. Return to work is often frustrating and characterized by reduced hours or adapted roles, anxiety about potential mistakes in cognitively demanding tasks, and lower self-confidence due to the inability to work weeks after infection [6,7]. The understanding and the definition of this phenomenon are still poor. Brain fog’s mechanisms may range from neuroinflammation to loss of grey matter, microvascular injury, brain stem dysfunction, a mild form of encephalopathy, disruption in perceptive systems, and psychiatric disorders [6,8,9,10]. Not to say, brain fog in different studies is included under different symptom categories. Some authors assimilate brain fog to neurologic symptoms [4,9,11,12,13]. Other studies include brain fog in cognitive symptom class [6,14,15], and finally, brain fog may be categorized among neuropsychiatric symptoms [10,16,17,18]. Despite such a high prevalence in the population, discrepancies in the definitions of brain fog are so frequent as to limit the accuracy of prevalence estimates and diagnostic criteria [13]. Far from being a trivial issue, this heterogeneity in classification may lead to a diagnostic underestimation and low efficacy in medical therapeutic approaches.

Moreover, patients with brain fog symptoms seeking medical consultation may experience further frustration due to healthcare professionals’ disbelief. In the absence of objective diagnostic criteria and evidence-rooted interventions for the follow-up [19], brain fog patients are frequently diagnosed as merely “long-haulers,” as depressed or anxious, or even dismissed as “being all in your head” [6,20,21]. Thus, despite severe functional impairment in daily activities, brain fog patients’ complaints may often be played down due to the reduced deepening of this phenomenon among healthcare professionals and the scarce consistency in definitions and etiopathogenetic hypotheses [15,22]. This poor focus on brain fog is also evident when we analyze its misty boundaries concerning other symptoms, namely fatigue and cognitive and psychiatric symptoms.

### 1.1. Brain Fog and Fatigue

Brain fog is sometimes described as part and parcel of chronic fatigue syndrome following the onset of infection. Chronic fatigue syndrome is another frequent post-infection sequela characterized by low energy and easiness of weakness even after light activities [18], making the individual perceive cognitive and motor tasks as extremely effortful [23,24]. In this perspective, brain fog would be one of the symptoms of this broader syndrome, best for cognitive aspects. On the other hand, several studies treat fatigue and brain fog as distinct post-infection sequelae, although frequently occurring together [2,13,25,26]. The former would involve mainly a sort of physical asthenia, while the latter would be characterized primarily by an excessive easiness to cognitive weakness. Thus, the role of brain fog as a symptom is still to be defined, but an additional issue requires clarification: whether brain fog may constitute a separate syndrome too. In particular, the association of brain fog with cognitive, psychiatric, and psychological factors is still quite misty.

### 1.2. Brain Fog and Cognitive Symptoms

The boundaries between brain fog and cognitive symptoms are pretty controversial in the literature. Several descriptions of brain fog variably cited problems in concentrating, disorientation and difficulty finding the right words [14], impairment in domains of attention, processing speed, and recalling information [15], worsening of working memory [4], confusion, and dizziness [27]. Moreover, difficulty with executive functioning, problem-solving, and decision-making, both short-term and long-term memory loss, were described [16]. In these studies, brain fog and forms of cognitive impairment would substantially overlap [4,6,14,17,23]. On the other hand, several studies keep brain fog, and cognitive deficits separate, suggesting that they represent two distinct clinical entities [7,9,10,13,16,26,28,29]. Despite this, data on their co-occurrence are lacking. Thus, the relationship between cognitive impairment and brain fog is still blurred, as the former might represent a primary effect of Long-COVID and, as such, be independent of, although frequently occurring with, brain fog. Otherwise, brain fog could be conceptualized as more substantial than the mere sensation of mental confusion but rather as a multilevel consequence of the infection, implying cognitive difficulties. Unfortunately, at the moment, both of the hypotheses are mere speculations.

### 1.3. Brain Fog and Neuropsychiatric Symptoms

Some studies suggested that long-haulers’ brain fog and cognitive impairment would be secondary to neuropsychiatric disorders. In other words, psychiatric and psychological symptoms, premorbid or beginning even at the acute stage of the infection, would negatively affect cognitive function, thus determining brain fog [30]. This perspective relies on cognitive failures’ relevant and diagnostic role in depressive and anxiety states [31]. In particular, learning and memory, as well as attention and concentration, are significantly impaired during depressive episodes [32]. Analogously, rumination and difficulties in decision-making are just as typical of depression, too [33,34]. However, the opposite hypothesis, that is, that self-appraisal of cognitive inefficiency may determine low mood and frustration, is equally valid. In a small study [8], cognitive complaints were associated with increased anxiety and depression, suggesting that patients are experiencing distress because they realize cognitive symptoms. In addition, sleep disturbances are pretty frequent in post-COVID patients, which may jeopardize mood and cognitive functioning [35,36]. Hence, each symptom’s primary or secondary role is yet to be proven.

To note, worldwide being a woman and lower age represented risk factors for developing psychological symptoms, such as anxiety, depression, and sleep disorders, due to the impact of the pandemic [35,37,38,39]. Analyzing Long-COVID data, the prevalence of the female gender is confirmed, while older subjects, rather than younger individuals, seem more vulnerable [40,41]. However, no data about demographic variables were explicitly highlighted concerning brain fog.

### 1.4. Aims and Hypotheses of the Study

Since brain fog has been defined as one of the most debilitating post-COVID manifestations leading to a decreased quality of life [16], a standardized definition and clinical understanding of this phenomenon are strongly required. The main aims of this study were to outline (a) the relationship between brain fog and cognitive symptoms, (b) the relationship between brain fog and neuropsychiatric symptoms, and (c) the role of demographic variables, i.e., gender and age, as related factors of brain fog symptomatology. We have developed a cross-sectional study that distributed a multidimensional web survey on an extensive adult Italian sample. We hypothesized that: (1) brain fog would be mainly co-occurring with subjective cognitive complaints, (2) subjects with brain fog would report higher levels of depression, anxiety, rumination, and sleep disorders, and (3) women and older subjects would show a higher probability of having brain fog.

## 2. Materials and Methods

### 2.1. Study Design

A cross-sectional web-based survey design was adopted. Employees of a large Italian banking group were invited to participate. All participants were provided with a detailed description of the experimental procedures and required consent before participating in the study. Participation was voluntary and not rewarded in any form. The survey was anonymous, as each participant was assigned an alphanumeric code to ensure the confidentiality of information. Each subject could fill out the questionnaire only once, although participants could terminate the survey within a week. Questionnaires were evenly distributed across the national territory. The study was conducted in accordance with the Declaration of Helsinki and under research protocols approved by local Ethical Committees (Scuola Normale Superiore and Scuola Superiore Sant’Anna Joint Ethical Committee: Protocol No. 04/2021).

### 2.2. Participants

An initial sample of 2312 employees of a large Italian banking group was invited to participate in the online survey. The research participation was voluntary and anonymous, and the inclusion criteria were: (a) age greater than or equal to 18 years, (b) Italian mother tongue or high-level knowledge of Italian language, and (c) living in Italy since the pandemic outbreak (i.e., since March 2020). We did not consider the subjects’ actual previous infection due to COVID-19 as an inclusion criterion, given the web-based design of the research and the consequent inability to collect reliable information about this point. Many people who experienced mild or asymptomatic SARS-CoV-2 infections did not receive antigen testing at the time of acute infection [42].

Not to say, some long haulers might not have detectable antibodies to SARS-CoV-2 when the first serological test became available commercially [9]. All these considerations prevented an undoubtful attribution of later symptoms to SARS-CoV-2 infection.

Nonetheless, some further remarks partially contain the relevance of this issue. Recent studies highlighted that post-COVID symptoms were not necessarily related to the severity of infection [43] and that similar long-term consequences were present in affected and not affected subjects [9,44,45]. Moreover, several studies on Long-COVID do not include a control group of uninfected subjects, thus preventing inferences about attributable risk [42].

### 2.3. Assessment

#### 2.3.1. Brain Fog and Subjective Cognitive Complaints

The phenomenon of brain fog was addressed by a specific question “Since the COVID-19 pandemic outburst to date, did you ever experience the onset or the worsening of the sensation of brain fog?” (Italian version: “Dall’inizio della pandemia da COVID-19 ad oggi, hai notato la comparsa o l’accentuarsi della sensazione di “mente annebbiata”?) The subject could answer on a four-point Likert scale (0 = No, never/Just like before the pandemic; 1 = Sometimes; 2 = Frequently; 3 = Mostly/Always).

Considering the web-based design of the research and thus the consequent inability to collect objective testing data, we explicitly focused on subjective cognitive complaints (Scc). Scc were investigated by three specific questions, namely (a) one about concentration: “Since the COVID-19 pandemic outburst to date, did you ever experience the onset or the worsening of troubles concentrating?” (Italian version: “Dall’inizio della pandemia da COVID-19 ad oggi, hai notato la comparsa o l’accentuarsi della difficoltà a concentrarti?”), (b) one about sustained attention: “Since the COVID-19 pandemic outburst to date, did you ever experience the onset or the worsening of difficulties in keeping your attention focused for long times?” (Italian version: “Dall’inizio della pandemia da COVID-19 ad oggi, hai notato la comparsa o l’accentuarsi della difficoltà a mantenere l’attenzione per un tempo prolungato”?), and (c) one about memory: “Since the COVID-19 pandemic outburst to date, did you ever experience the onset or the worsening of memory difficulties?” (Italian version: “Dall’inizio della pandemia da COVID-19 ad oggi, hai notato la comparsa o l’accentuarsi della difficoltà a memorizzare le informazioni?”). For each question, a four-point Likert scale (0 = No, never/Just like before the pandemic; 1 = Sometimes; 2 = Frequently; 3 = Mostly/Always) was given to answer.

We included each subject of the sample in one and only one of the following three groups: (a) Healthy Subjects (HS), defined as subjects who reported neither brain fog sensation nor Scc. Thus, we included subjects in this group if they scored 0 on the brain fog item and all the three Scc items; (b) Probable Brain Fog (PBF), that is, subjects who reported only brain fog sensation but not Scc, or alternatively, subjects who reported Scc without brain fog sensation. Thus, we included subjects in the PBF group if they scored equal/higher than 1 on the brain fog item but 0 in all the three Scc items, or otherwise, if they scored equal/higher than 1 in at least one of the three Scc items but 0 on the brain fog item; (c) Functional Brain Fog (FBF) included subjects who reported brain fog sensation associated with Scc. Thus, we included subjects in the FBF group if they scored equal/higher than 1 on the brain fog item and equal/higher than 1 in at least one of the three Scc items.

Furthermore, to consider FBF severity, we considered the four items of brain fog and Scc as constituting a single questionnaire scoring from a minimum of 2 (mild severity) to a maximum of 12 (high severity).

#### 2.3.2. Positive Affect and Negative Affect Schedule (PANAS)

Participants’ affect was recorded by a validated Italian version of the PANAS [46,47]. It is a 20-item self-report measure whose scores are rated on a 5-point Likert scale (1 = never; 5 = always). The questionnaire investigates two independent affective dimensions, as half of the items constitute the Positive Affect subscale, whereas the remaining half constitutes the Negative Affect subscale. The Positive Affect scale reflects how much a person feels enthusiastic, excited, active, and determined. The Negative Affect scale reflects a general dimension of unpleasant engagement, including a broad range of aversive affects, such as fear, nervousness, guilt, and shame.

#### 2.3.3. Depression, Anxiety and Stress Scale (DASS-21)

Perceived depression, anxiety, and stress were assessed by the Italian version of the DASS-21 [48,49]. It is a 21-item self-report questionnaire assessing core anxiety, depression, and stress symptoms. Each item is scored on a 4-point Likert scale (0 = “Never”; 3 = “Always”). The DASS-21 has shown good psychometric properties (i.e., internal/external consistency and validity, test-retest stability) in clinical and non-clinical samples [49]. It contains three subscales: Depression, Anxiety, and Stress.

#### 2.3.4. Impact of Event Scale-Revised (IES-R)

The IES-R [50] was used to assess post-traumatic symptoms in a non-clinical setting. The IES-R is a self-report measure of current subjective distress in response to a traumatic event. According to the Diagnostic and Statistical Manual of mental disorders (DSM-IV) [51] criteria for post-traumatic stress disorder (PTSD), the IES-R comprises three classes of symptoms: intrusive recollections of the traumatic event, avoidant symptoms, and hyperarousal symptoms, thus resulting in three subscales (Intrusion, Avoidance, and Hyperarousal, respectively) representative of the primary symptom clusters of PTSD. The IES-R is a self-report questionnaire that consists of 22 items, and each is scored on a 5-point Likert scale (0 = Not at all; 4 = Extremely). Moreover, the IES-R provides a cut-off score (≥33), highlighting phenomena worth clinical attention [52]. Our study adopted a validated Italian version [53].

#### 2.3.5. Beck Cognitive Insight Scale

The Beck Cognitive Insight Scale (BCIS) [54,55] is a standardized self-rated instrument composed of 15 items that examine different dimensions of self-reflection. The BCIS relies on two sub-factors: Self-Reflectiveness and Self-Certainty. Self-Reflectiveness concerns the willingness to acknowledge fallibility, self-appraise, and consider external feedback, while Self-Certainty expresses confidence in one’s judgments and opinions. A subscale of nine items assesses Self-Reflectiveness (e.g., “If somebody points out that my beliefs are wrong, I am willing to consider it”), while a subscale of six items assesses Self-Certainty (e.g., “I can trust my judgment at all times”). A third index, R-C, resulting from the Self-Reflectiveness score minus the Self-Certainty score, reflects the balance between the two components. Developed initially to investigate insight in individuals with psychotic disorders, the BCIS showed that in healthy subjects, higher levels of Self-Reflectiveness are related to higher rumination, indecision, and finally, lower self-confidence in one’s decision-making ability. In turn, this jeopardizes the decision-making function [56].

#### 2.3.6. Sleep Disorders

The following question investigated sleep disorders: “At the moment, how would you describe the quality of your sleep?” Italian version: “Al momento, come giudicheresti la qualità del tuo sonno?”). Participants could answer choosing between four alternatives: (a) “My sleep is regular and satisfying”, (b) “I find it hard to fall asleep in the evening”, (c) “I wake up too early in the morning without any reason”, and (d) “My sleep is quite disturbed, and I wake up repeatedly during the night without any necessity”. If participants answered (a), they were categorized as not having sleep disorders. Otherwise, choosing one of the three remaining options, they were classified as having sleep disorders.

### 2.4. Statistical Analyses

Statistical analyses were performed using IBM SPSS software ver. 26.

Comparisons between HS, PBF, and FBF groups on nominal variables, namely gender and sleep disorders, were made using the chi-square test. A one-way ANOVA was performed for the continuous variable age, followed by Bonferroni’s post hoc test.

Comparisons between the three groups on continuous variables, namely BCIS subscales (Self-Reflectiveness, Self-Certainty, and R-C), PANAS subscales (Negative Affect and Positive Affect), IES-R subscales (Avoidance, Intrusion, and Hyperarousal) and total score, and DASS-21 subscales (Stress, Anxiety, and Depression) and total score, were performed by a one-way analysis of covariance (ANCOVA) followed by Bonferroni’s post hoc tests, controlling for gender (covariate).

A stepwise multiple regression investigated the relationship between the FBF severity score (dependent variable) and demographic and neuropsychiatric factors (independent variables) in the FBF group model. Pre-selection of independent variables to include in each regression model was carried out using Spearman’s correlation analysis in the FBF group. The FBF severity score was considered as the dependent variable and age, gender, DASS-21 total score, IES-R total score, PANAS negative affect subscale, and BCIS Self-Reflectiveness as independent variables. Only variables with *p* < 0.05 in the pre-selection correlation analyses were entered as independent variables in the regression models.

## 3. Results

The final sample consisted of 441 subjects. According to the criteria previously pointed out, the HS group included 147 subjects (33%), the PBF group 117 subjects (27%), and the FBF group 177 subjects (40%). The socio-demographic characteristics of the three groups are described in Table 1. The chi squared analyses showed that the three groups differed in gender and prevalence of sleep disorders. No significant differences between the three groups emerged with regard to age as highlighted by the one-way ANOVA analysis. The results for demographic and neuropsychiatric in the three groups are described in Table 1.

### 3.1. Neuropsychiatric Characteristics of HS, PBF, and FBF Groups

The one-way ANCOVA showed that the BCIS Self-Reflectiveness level was significantly different in the three groups (F**_2.437_**, 5.825; *p* = 0.003) while controlling for gender (covariate); Bonferroni’s post hoc highlighted that there was a statistically significant difference between HS and FBF (*p* = 0.003), stressing the highest level in the FBF group. No significant difference between groups for the Self-Certainty and the R-C indices emerged.

In the PANAS questionnaire, the Positive Affect level was significantly different in the three groups (F_2.435_, 12.897; *p* < 0.001) while controlling for the covariate. Bonferroni post hoc test for multiple comparisons highlighted a statistically significant difference between the HS and PBF groups (*p* = 0.003) and between the HS and FBF groups (*p* < 0.001), thus stressing in both cases the highest level in the HS group. There was no statistically significant difference between PBF and FBF groups (Figure 1a). The PANAS Negative Affect level was statistically different in the three groups (F_2.435_, 47.460; *p* < 0.001) while controlling for the covariate. Bonferroni’s post hoc test highlighted that there was a statistically significant difference between the HS and PBF groups (*p* = 0.003), the HS and the FBF groups (*p* < 0.001), and the PBF and FBF groups (*p* <0.001), stressing the highest level in the FBF group (Figure 1b).

In the IES-R scale, the total score was statistically different in the three groups (F**_2.434_**, 42.602; *p* < 0.001) while controlling for the covariate. Bonferroni’s post hoc test highlighted a statistically significant difference between the HS and the PBF groups (*p* = 0.024), the HS and FBF groups (*p* < 0.001), and the PBF and FBF groups (*p* < 0.001), stressing the highest level in the FBF group (Figure 1c). The Avoidance sub-score was significantly different in the three groups (F_2.434_, 36.082; *p* < 0.001) while controlling for the covariate. Bonferroni post hoc test for multiple comparisons highlighted a statistically significant difference between the HS and the FBF groups (*p* < 0.001) and between PBF and the FBF groups (*p* < 0.001), stressing in both cases the highest level in the FBF group. Likewise, the Intrusion sub-score was statistically different in the three groups (F_2.434_, 28.843; *p* < 0.001) while controlling for the covariate. Bonferroni’s post hoc test highlighted a statistically significant difference between the HS and the FBF groups (*p* < 0.001) and the PBF and the FBF groups (*p* < 0.001), stressing in both comparisons the highest level in the FBF group. Finally, the IES-R Hyperarousal sub-score was statistically different in the three groups (F_2.434_, 41.360; *p* < 0.001) while controlling for the covariate. Bonferroni’s post hoc test highlighted a statistically significant difference between the HS and the PBF groups (*p* < 0.040), the HS and the FBF groups (*p* < 0.001), and the PBF and FBF groups (*p* < 0.001), stressing the highest level in the FBF group.

Concerning the DASS-21 questionnaire, the total score was statistically different in the three groups (F_2.435_, 44.394; *p* < 0.001) while controlling for the covariate. Bonferroni’s post hoc test highlighted a statistically significant difference between the HS and the PBF groups (*p* = 0.002), the HS and FBF groups (*p* < 001), and the PBF and FBF groups (*p* < 0.001), stressing the highest level in the FBF one (Figure 1d). In the Depression Anxiety Stress Scales-21 (DASS-21), the Stress level was significantly different in the three groups (F_2.435_, 52.713; *p* < 0.001) while controlling for the covariate. Bonferroni post hoc test for multiple comparisons highlighted a statistically significant difference between the HS and PBF groups (*p* < 0.001), the HS and FBF groups (*p* <0.001), and the PBF and FBF groups (*p* < 0.001), stressing the highest level in the FBF one. The DASS-21 Anxiety sub-score was statistically different in the three groups (F_2.435_, 21.692; *p* < 0.001) while controlling for the covariate. Bonferroni’s post hoc test highlighted a statistically significant difference between the HS and the FBF groups (*p* < 0.001) and the PBF and FBF groups (*p* < 0.001), stressing the highest level in the FBF one. The DASS-21 Depression sub-score was statistically different in the three groups (F_2.435_, 31.718; *p* < 0.001) while controlling for the covariate. Bonferroni’s post hoc test highlighted a statistically significant difference between the HS and the PBF groups (*p* = 0.038), the HS and FBF groups (*p* < 001), and the PBF and FBF groups (*p* < 0.001), stressing the highest level in the FBF one.

### 3.2. Correlates and Predictors of Brain Fog and Cognitive Complaints in Functional Brain Fog Group

In the FBF group, the brain fog plus Scc total score was positively correlated to gender (r = 0.193, *p* = 0.009), sleep disorders (r = 0.215, *p* = 0.004), the DASS-21 total score (r = 0.443, *p* <0.001), the IES-R total score (r = 0.324, *p* < 0.001) and the PANAS Negative Affect score (r = 0.312, *p* < 0.001) (Table 2).

The multiple stepwise regression final model was significant (F_3.175_, 28.966; *p* < 0.001). The DASS-21 total score was selected as the most influential predictor of brain fog plus the Scc score in the first step (R^2^ = 0.278). In the second step, gender and the DASS-21 total score were selected as significant predictors (R^2^ = 0.311). In the third step, gender, the DASS-21 total score, and the IES-R total score were selected as significant predictors (R^2^ = 0.336) (Table 3).

## 4. Discussion

The main aim of this study was to deepen the characteristics of the post-COVID brain fog phenomenon and, in particular, its relationship with cognitive and neuropsychiatric symptoms. As a secondary purpose, we aimed to investigate the role of socio-demographic variables as related factors of brain fog symptomatology. Three main results emerged. First, as expected, brain fog mainly co-occurred with Scc, thus resulting in a brain fog disorder (FBF), representing 40% of respondents. However, brain fog and Scc occurred separately (PBF) in some cases. Second, psychiatric symptoms were substantially involved in FBF. Third, female gender but not age predicted a higher vulnerability to developing FBF.

### 4.1. Brain Fog and Scc

As hypothesized, brain fog and Scc were strictly associated. Moreover, FBF differed from the other two groups for the highest frequency of sleep disturbances and the highest levels of psychopathological symptoms. Moreover, the FBF group was characterized by higher levels of ruminative attitude, as mirrored by SR scores [56,57]. Previous studies showed that cognitive insight and Scc are close constructs, and specifically that the greater the Scc, the higher the SR [58,59]. Thus, FBF showed lower confidence in one’s cognitive abilities related to the greater tendency to rely on external support, rumination, indecision, and, therefore, less efficient decision-making function or depressive mood. We may hypothesize that subjects more prone to more significant uncertainty are also more likely to develop depression, anxiety, subjective cognitive overload, and brain fog. Otherwise, we may speculate that FBF may facilitate low self-confidence in one’s decision-making ability.

The PBF group consisted of subjects who described only brain fog or cognitive failures and showed higher psychopathological symptoms, except for sleep disturbances and rumination/indecision, than the HS group. Conversely, the PBF group showed a lower prevalence (27%) and significantly lower levels of all psychological and neuropsychiatric symptoms than the FBF group. Specifically, subjects reporting only cognitive failures represented 22%, and subjects reporting only brain fog were 5%. This evidence suggested a sort of continuum of the brain fog phenomenon, characterized by a progressive worsening of quality of life and psychological well-being, with PBF as a borderline between HS and FBF and FBF as the actual brain fog disorder. Interestingly, our data point to brain fog as a syndrome associated with Scc and brain fog as a symptom, which may stand alone or characterize other disorders, such as chronic fatigue syndrome. Furthermore, our data suggest that brain fog sensation and Scc do not necessarily coincide, are not necessarily associated, and are not interchangeable.

Our study contributes to classify by explicit and clear criteria the post-COVID brain fog phenomenon. Significant inconsistencies characterize prevalence estimates of brain fog in the literature. Few studies reported brain fog prevalence separately, and data may range from 2% to 10% [60,61]. However, several studies described difficulties that might resemble brain fog, but under different denominations, such as difficulties in daily activities [62,63], mental slowness [64], mental fog [60,65], difficulty or problems with thinking [66,67]. This variability in naming corresponds to great variability in prevalence data, ranging from 10% to 68%. Even more remarkable is the gap, between 22% and 88%, when brain fog is included in the list of post-COVID cognitive impairments [2,16]. Again, brain fog can be listed apart from cognitive failures but assimilated into attention disorders [68]. Not to mention those studies that report attention/concentration difficulties but do not explicitly cite brain fog [26]. These inconsistencies in definitions make epidemiological comparisons quite disputable. Likewise, similar biases about data on Long-COVID cognitive sequelae emerge comparing results of reviews and meta-analyses. In this case, a tremendous variability in data ranging from 20 to 88% can be detected [16,69], depending on the specific cognitive function considered. Indeed, some previous studies have hinted at a strict relationship between cognitive impairments and brain fog, although, differently from our approach, failing to provide a veritable model and consistent data [6,12,30].

Post-mortem testing on patients affected by COVID-19 revealed multifocal vascular damages, even in absence of underlying risk factors [70]. The primary transmission modes of the SARS-CoV-2 are contact, droplet, and airborne. In particular, airborne transmission implies respiratory droplets which may get deposited deep into the respiratory tract [71] and eventually could enter the brain via the olfactory nerve tract and eventually enter the brain via the olfactory nerve tract. Then, they may reach right temporal lobe, limbic and paralimbic regions, the cerebellum, and the hypothalamus [10,13,72]. In particular, in the median eminence of the hypothalamus, there is a high concentration of mast cells and microglia, which are vulnerable to viruses such as SARS-CoV-2 and release pro-inflammatory molecules. Thus, they would contribute to brain inflammation and, finally, brain fog and/or cognitive dysfunction [10,73].

### 4.2. The Role of Neuropsychiatric Symptoms

As expected, our second result showed significant involvement of neuropsychiatric symptoms in the brain fog disorder. Firstly, comparisons between the three groups showed apparent differences in the co-occurrence of low mood and adverse affective reactions, sleep disorders, and rumination/indecision. The worst psychological adaption emerged in the FBF group. Moreover, correlation analysis stressed that in the FBF group, the higher the psychological distress, the greater the cognitive burden and sense of blunting. Finally, depression, anxiety, and post-traumatic reaction emerged as predictors of the severity of brain fog in the FBF group. Not to say, the greatest odds of subjects scoring above the critical thresholds at the IES-R and the DASS-21 (15% and 24%, respectively) were in the FBF group. This evidence supports a significant involvement of neuropsychiatric factors in the brain fog syndrome and warns that they should be considered in particular when approaching a patient with brain fog [74].

Indeed, neuropsychiatric symptoms are frequently disclosed as Long-COVID sequela. On the one hand, self-appraisal of cognitive difficulties likewise triggers depression [1,7,75,76]. On the other hand, depression usually interferes with cognition to the extent that poor concentration is a common diagnostic criterion for many mood and anxiety disorders [77]. Moreover, depression may determine a hyperawareness of cognitive [78]. Thus, subjective experience of brain fog and cognitive failures can be generated, exacerbated, or perpetuated by stress, depression, or anxiety, independently of an organic cause [6,9].

Among the psychological models, several studies focused on post-traumatic symptoms [79,80,81], and sleep disturbances, another typical Long-COVID symptom [13,35,39].

Moreover, some personality traits may play a role in facilitating the onset or the worsening of depression and anxiety [13], which in turn may have triggered neuropsychological mist. Patients with post-COVID symptoms showed higher levels of neuroticism, and lower scores for positive mood and self-control [82].

Thus, consistently with psychological models, we might speculate that FBF and PBF could develop without the infective chain of events. In our study, the rate of FBF and PBF cases amounts to 65%, and we had no information about previous COVID-19 infections in our sample. We cannot exclude a share of not infected subjects included in the FBF and PBF groups. The harshly traumatic experience represented by the social isolation, the unexpected and sudden threat to one’s health and the health of the loved ones, the uncertainty also due to the repeated challenges of several variants of the virus [83], and the ongoing adjustment to new private and working life habits [84] might determine similar effects than the organic etiopathogenic course.

### 4.3. The Role of Demographic Variables

As expected, our third result showed that the female gender was a risk factor for developing the FBF syndrome. This result is consistent with previous studies reporting that females were more prone to develop post-COVID symptomatology, including cognitive difficulties [2,28,85,86,87]. Different autoimmune mechanisms and pro-inflammatory reactions are among the most cited explanations [9,88]. Other authors suggest a strict relationship between female gender prevalence and depressive/anxiety symptoms [28]. In the literature, there is consensus on the greater vulnerability to psychological distress due to the pandemic events in women than in men. Another possible explanation is the greater willingness of women to report their own psychological and affective states than men [81].

Against expectations, brain fog phenomena did not significantly imply age. This result contrasts both with the literature’s frequent evidence that younger people are at higher risk of psychological maladaptation and the previous studies on post-COVID symptoms. Indeed, data on the role of age in Long-COVID and brain fog are not that consistent in the literature. Some studies described older age as a risk factor, with a higher frequency of post-COVID symptoms and cognitive impairments [2,7,41]. In other studies, such association did not emerge [40,60]. This lack of evidence regarding age may be traced back to the biochemical component of FBF, which would be relatively independent of age. However, further studies on this point are required.

### 4.4. A Bio-Psycho-Social Model of Brain Fog

Our data encourage us to overcome the mere polarization between organic and psychosocial factors. As previously stated, our results underline a significant relationship between FBF and stress and post-traumatic symptoms. As a complement to the psychological backlash of the pandemic events, the effects of SARS-CoV-2 infection may manifest as altered stress responses.

Chronic activation of the extended autonomic system, including the neuroendocrine and neuroimmune systems, would be associated with an increased risk of developing Long-COVID [29,89]. As mentioned before, SARS-CoV-2 could reach the hypothalamus and activate brain mast cells and microglia to release pro-inflammatory molecules [3,10]. In turn, the autonomic alteration of hypothalamus function may determine cognitive abnormality, sleep dysregulation, and profound fatigue. Moreover, the involvement of the limbic system may jeopardize affective regulation [86].

Further data showed that smaller hippocampal, left amygdala, and anterior cingulate cortex volumes characterized subjects affected by post-traumatic symptoms or even only exposed to traumatic events than controls [90,91]. Another study showed that PTSD patients, compared to controls, were characterized by weaker positive connectivities between the middle prefrontal cortex and the amygdala, hippocampus, parahippocampal gyrus, and rectus as between the inferior orbitofrontal cortex and the hippocampus [92]. Thus, altered emotional responsiveness can interfere with the recruitment of regions implicated in top-down attentional control, thus accounting for attentional biases, memory, and working memory failures [93,94]. Finally, dysfunction of brain areas such as the right temporal lobe, the amygdala, the hippocampus, the brainstem, the cerebellum, and the hypothalamus might underpin cognitive and psychiatric symptoms due to COVID-19 infection [72,74]. In light of this evidence, our data suggest that the heavily stressful condition determined by COVID-19 pandemic, interacting or not with biochemical events due to the infection, might generate such a psychological burden to affect mood and neuropsychological function [95].

### 4.5. Limitations of the Study

We must consider some limitations when interpreting the results of the present study. First, our main concern is selection bias. We recruited only employees of a single Italian banking group, so the results may not represent the general Italian population. Moreover, those who have accepted our invitation to participate in the study as volunteers may also be more engaged with the topic and more sensitive toward psychological issues. However, the width of the sample allows us to cautiously speculate about the possibility of extending our results to more significant segments of the Italian population.

Second, we investigated brain fog and Scc by a single-item question. Thus, possibly they may not fully catch the multifaceted phenomenon object of the study. Moreover, neuropsychological testing could not ascertain cognitive symptoms but somewhat subjective reports. Thus, the lack of objective memory tests limits firm conclusions. This limitation is expected to be faced by any web-based study which precludes features of the neurologic exam, thus potentially introducing an information bias [9,40,67].

Nonetheless, several studies state that self-appraisal of cognitive failures may indicate even subtle changes in cognitive function, and Scc may hint at objective problems to the extent that in the elderly self-reported cognitive impairments are considered a risk factor for mild cognitive impairment or dementia [78,96]. However, a more objective neuropsychological examination is warranted in future studies.

Third, our study did not document the subjects’ premorbid cognitive and psychiatric status. However, the questions about brain fog and cognitive symptoms required an onset or a worsening following the COVID-19 pandemic outburst in Italy (i.e., March 2020). This criterion would allow us to infer that the COVID-19 pandemic could account for the symptoms described.

Fourth, as widely stressed above, we have no information about previous COVID-19 infections in our sample. Despite this, in our study, the post-pandemic outburst criterion undoubtfully relates FBF and PBF to pandemic events, in terms of infection and/or psychologically traumatic consequences. However, this speculation requires future accurate deepening. Further studies comparing a group of confirmed infected patients and a control group including uninfected subjects could better cast light on our results and deepen our conclusions on FBF and PBF.

## 5. Conclusions

Post-COVID-19 syndrome, particularly FBF, poses a significant long-term global public health concern. It can be very debilitating and may even jeopardize economic consequences for individuals, their families, and society as a whole [97].

Many different types of viruses [18] and even treatments affecting the immune system, such as chemotherapy for cancer [98], can cause long-term illness, and disorders similar to brain fog.

Our attempt to propose a systematic conception of post-COVID-19 brain fog would contribute to develop consistent diagnostic criteria [13,25,99], and facilitate assessment, differential diagnosis, and treatment [15].

Our study took a cue from the several reviews and meta-analyses which classified brain fog either as a neuropsychiatric, cognitive or affective phenomenon, and variously separated brain fog from cognitive difficulties. Thus, by a systematic approach, we attempted to verify whether this distinction was valid or not. Indeed, our results showed that most cases of what we call “brain fog” in long haulers are characterized by the association of brain fog and cognitive difficulties (FBF). Nonetheless, we cannot disregard a minority of subjects who describe either brain fog or cognitive failures (a condition not casually named “Probable” Brain Fog). Furthermore, we suggest a biopsychosocial framework for brain fog characterized by a dysfunction of brain-body circuits and networks that interact with a potential triggering event and psychological factors [100]. Nonetheless, without brain imaging or biochemical data, caution is required to infer a definite etiology of FBF [14].

In conclusion, more profound knowledge of FBF could lead to better long-term treatments and rehabilitation, thus possibly contributing to reducing this pandemic’s long-term health and socioeconomic burden [12]. Healthcare practitioners, better aware of the incidence, features, and implications of COVID-19 FBF, may refer patients for appropriate assessment and consider specific treatment options for managing the symptoms [4,101]. On the other hand, public resource allocation would prioritize diagnostics, rehabilitation, and psychological support [95].

## Figures and Tables

**Figure 1 jcm-11-05529-f001:**
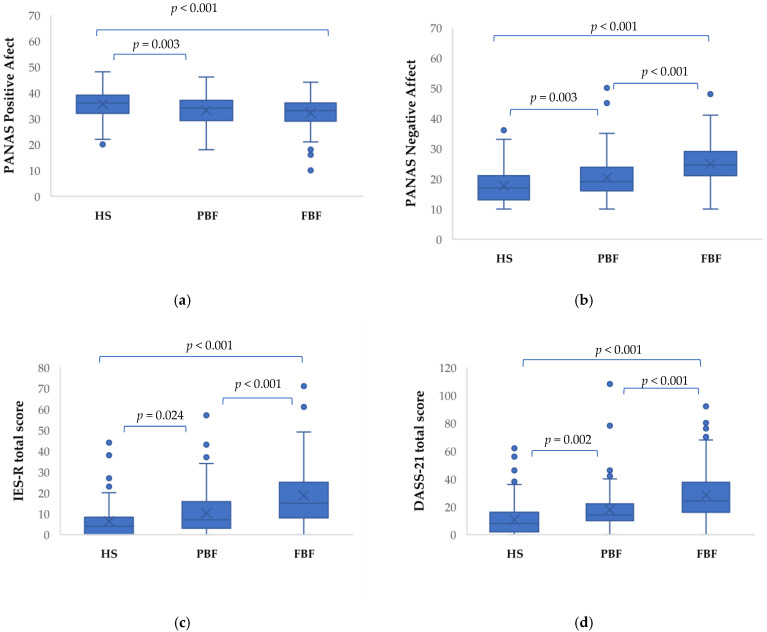
Boxplots and *p*-values of ANCOVA tests comparing HS, PBF and FBF on neuropsychiatric characteristics. PBF and FBF showed higher levels of Negative Affect (**b**), post-traumatic symptoms (**c**) and psychopathological symptoms (Fig.1d). HS showed higher levels than PBF and FBF only for Positive Affect (**a**). FBF showed higher levels than PBF of Negative Affect (**b**), post-traumatic and psychopathological symptoms (**c**,**d**). HS = Healthy Subjects; PBF: Probable Brain Fog FBF: Functional Brain Fog; IES-R = Impact of Event Scale-Revised; DASS-21 = Depression Anxiety Stress Scales-21; PANAS = Positive Affect and Negative Affect Schedules; PA = Positive Affect; NA = Negative Affect.

**Table 1 jcm-11-05529-t001:** Demographic and neuropsychiatric characteristics of Healthy Subjects, Probable Brain Fog, and Functional Brain Fog.

Variable	HS *n*= 147	PBF *n* = 117	FBF *n* = 177	*p*	HS vs. PBF		HS vs. FBF		PBF vs. FBF	
					Crit. Diff.	*p*	Crit. Diff.	*p*	Crit. Diff.	*p*
Females *n* (%)	39 (26.5)	43 (36.8)	89 (50.3)	**<0.001**	-	-	-	-	-	-
Age (years ± SD)	47.84 ± 9.6	47.93 ± 10.0	45.72 ± 8.8	0.063	-	-	-	-	-	-
Sleep disorders Yes n (%)	43 (29.3)	55 (47)	134 (75.7)	**<0.001**	-	-	-	-	-	-
BCIS Self-Reflectiveness (mean ± SD)	9.67 ± 3.1	10.56 ± 3.0	10.9 ± 3.2	**0.003**	−0.876	0.075	−1.197	**0.003**	−0.321	1.000
BCIS Self-Certainty (mean ± SD)	7.88 ± 2.9	8.45 ± 2.6	8.40 ± 2.7	0.132	-	-	-	-	-	-
BCIS R-C Index (mean ± SD)	1.8 ± 4.2	2.11 ± 4.2	2.51 ± 4.50	0.430	-	-	-	-	-	-
PANAS Positive Affect (mean ± SD)	35.6 ± 5.7	33.16 ± 5.8	32.17 ± 5.65	**<0.001**	2.361	**0.003**	3.245	**<0.001**	0.884	0.595
PANAS Negative Affect (mean ± SD)	17.63 ± 5.6	20.36 ± 6.6	24.87 ± 6.7	**<0.001**	−2.574	**0.003**	−6.882	**<0.001**	−4.308	**<0.001**
IES-R Avoidance (mean ± SD)	0.30 ± 0.4	0.50 ± 0.5	0.83 ± 0.65	**<0.001**	−0.165	0.053	−0.523	**<0.001**	−0.358	**<0.001**
IES-R Intrusion (mean ± SD)	0.29 ± 0.4	0.48 ± 0.6	0.82 ± 0.7	**<0.001**	−0.180	0.054	−0.517	**<0.001**	−0.337	**<0.001**
IES-R Hyperarousal (mean ± SD)	0.25 ± 0.4	0.44 ± 0.5	0.88 ± 0.8	**<0.001**	−0.191	**0.04**	−0.621	**<0.001**	−0.430	**<0.001**
IES-R Total score (mean ± SD)	6.33 ± 8.7	10.32 ± 10.5	18.57 ± 12.8	**<0.001**	−3.904	**0.024**	−12.047	**<0.001**	−8.142	**<0.001**
DASS-21 Stress (mean ± SD)	2.50 ± 2.6	4.32 ± 3.1	6.12 ± 3.2	**<0.001**	−1.772	**<0.001**	−3.524	**<0.001**	−1.752	**<0.001**
DASS-21 Anxiety (mean ± SD)	0.84 ± 1.7	1.43 ± 2.3	2.82 ± 3.3	**<0.001**	−0.553	0.265	−1.890	**<0.001**	−1.337	**<0.001**
DASS-21 Depression (mean ± SD)	1.93 ± 2.5	3.09 ± 3.2	5.19 ± 4.2	**<0.001**	−1.067	**0.038**	−3.048	**<0.001**	−1.981	**<0.001**
DASS-21 Total score (mean ± SD)	10.54 ± 11.8	17.68 ± 15.1	28.27 ± 19.1	**<0.001**	−6.784	**0.002**	−16.923	**<0.001**	−10.139	**<0.001**

HS = Healthy Subjects; PBF: Probable Brain Fog; FBF: Functional Brain Fog. BCIS = Beck Cognitive Insight Scale; PANAS = Positive Affect and Negative Affect Schedules; IES-R = Impact of Event Scale-Revised; DASS-21 = Depression Anxiety Stress Scales-21; SD = standard deviation; crit. diff. = critical difference. Significant values (*p* < 0.05) are highlighted in bold.

**Table 2 jcm-11-05529-t002:** Demographic and neuropsychiatric correlates of FBF severity.

Variable	FBF N = 177	
	r	*p*
Age	0.043	0.573
Gender	0.195	**0.009**
Sleep disorders	0.215	**0.004**
DASS-21 total score	0.443	**<0.001**
IES-R total score	0.324	**<0.001**
PANAS NA	0.312	**<0.001**
BCIS Self-Reflectiveness	0.095	0.208

FBF: Functional Brain Fog; IES-R = Impact of Event Scale-Revised; DASS-21 = Depression Anxiety Stress Scales-21; BCIS = Beck Cognitive Insight Scale; PANAS = Positive Affect and Negative Affect Schedules; NA = Negative Affect. Significant values (*p* < 0.05) are highlighted in bold.

**Table 3 jcm-11-05529-t003:** Predictors of FBF severity.

Variable		
	β (t)	95% CI
Step 1		
DASS-21 total score	**0.527 (8.179)**	[0.048, 0.078]
Gender	0.185 (2.914)	
IES-R total score	0.162 (2.218)	
Sleep disorders	0.127 (1.935)	
PANAS NA	−0.032 (−0.315)	
R^2^	0.278	
P	<0.001	
Step 2		
DASS-21 total score	**0.503 (7.910)**	[0.045, 0.075]
Gender	**0.185 (2.914)**	[0.314, 1.460]
IES-R total score	0.178 (2.501)	
Sleep disorders	0.101 (1.544)	
PANAS NA	−0.029 (−0.298)	
R^2^	0.311	
*p*	0.004	
Step 3		
DASS-21 total score	**0.415 (5.779)**	[0.033, 0.066]
Gender	**0.197 (3.136)**	[0.375, 1.506]
IES-R total score	**0.178 (2.501)**	[0.006, 0.051]
Sleep disorders	0.103 (1.605)	
PANAS NA	−0.077 (−0.786)	
R^2^	0.336	
*p*	0.013	

IES-R = Impact of Event Scale-Revised; DASS-21 = Depression Anxiety Stress Scales-21; PANAS = Positive Affect and Negative Affect Schedules; NA = Negative Affect. Significant values (*p* < 0.05) are highlighted in bold. Final significant predictors of FBF severity are: IES-R total score, DASS-21 total score and female gender.

## Data Availability

Not applicable.

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
