# Peer review of "A New Look on Long-COVID Effects: The Functional Brain Fog Syndrome"

_jcm, 2022, doi:10.3390/jcm11195529_

Round 1

Reviewer 1 Report

This articls is well written and contains good information,  some points need to be addressed before its publication as mentioned below:

1. Use Covid-19 or COVID-19 throughout the manuscript. There should be uniformity in the whole text. Update accordingly.

2. Please provide some information on the routes of transmission of SARS-CoV-2 in the introduction of the manuscript. Read & cite the paper mentioned below;

Aerosol transmission of SARS-CoV-2: The unresolved paradox. Trav Med Infect Dis 2020; 37:101869. https://10.1016/j.tmaid.2020.101869.

3. Cite the paper: Will the next spillover pandemic be deadlier than the COVID-19?: A wake-up call. Int J Surg. 97:106208.

4. In the text, the references should be in the number format. Please update accordingly.

5. Some references are not as per the format of the journal. Please arrange accordingly.

Rest is ok.

Author Response

We thank the reviewer for the positive comments and for highlighting some inaccuracies in the text and the bibliography. Furthermore, we have appreciated the bibliographical suggestions to refine the paper.  

This articls is well written and contains good information,  some points need to be addressed before its publication as mentioned below:

  1. Use Covid-19 or COVID-19 throughout the manuscript. There should be uniformity in the whole text. Update accordingly.

We thank the referee for the remark. Accordingly, we have opted for COVID-19 throughout the paper. Furthermore, we have replaced “long-Covid” with “Long-COVID”.

  1. Please provide some information on the routes of transmission of SARS-CoV-2 in the introduction of the manuscript. Read & cite the paper mentioned below;

Aerosol transmission of SARS-CoV-2: The unresolved paradox. Trav Med Infect Dis 2020; 37:101869. https://10.1016/j.tmaid.2020.101869.

We thank the referee for the suggestion according to which we have included the point not in the Introduction but rather in the Discussion to make more exhaustive the hypothetical etiopathogenic model for Scc and brain fog. The sentence has been rephrased as follows: “The primary transmission modes of the SARS-CoV-2 are contact, droplet, and airborne. In particular, airborne transmission implies respiratory droplets which may get deposited deep into the respiratory tract [71] and eventually could enter the brain via the olfactory nerve tract. Then, they may reach the right temporal lobe, limbic and para-limbic regions, the cerebellum, and the hypothalamus [10,13,72]. In particular, in the median eminence of the hypothalamus, there is a high concentration of mast cells and microglia, which are vulnerable to viruses such as SARS-CoV-2 and release pro-inflammatory molecules. Thus, they would contribute to brain inflammation and, finally, brain fog and/or cognitive dysfunction [10,73].” (page 14, lines 500-510).

  1. Cite the paper: Will the next spillover pandemic be deadlier than the COVID-19?: A wake-up call. Int J Surg. 97:106208.

We thank the referee for the suggestion. We have included the citation to make even more comprehensive the following paragraph: “The harshly traumatic experience represented by the social isolation, the unexpected and sudden threat to one’s health and to the health of the loved ones, the uncertainty due the repeated challenges of several variants of the virus [83] and the ongoing adjustment to new private and working life habits [84] might determine similar effects than the organic etiopathogenic course (page 15-16, lines 581-585)

  1. In the text, the references should be in the number format. Please update accordingly.

We thank the referee for the remark. We have corrected the residual typos throughout the paper.

  1. Some references are not as per the format of the journal. Please arrange accordingly.

We thank the referee for the note. We have performed a full revision of the references section.

Rest is ok.

Reviewer 2 Report

- Please try to spell out any abbreviation at its 1st time eg DSM-IV and PTSD

- In table 1: please delete the columns named "F (or chi-squared) and df". Please try to spell out any abbreviation in the table footnote.

Please try to report DASS-21 Total scores comparison of the 3 groups using boxplots and add overall p value inside the figure. Authors may do that with some other variables of their choice as well. Adding a graphical summary of their finding will be more attractive to the reader.

- Please expand the legend for table 3 and explain the results. It is better to report beta as a regression coeficient and report its 95%CI. Reader will not be interested the stepwise approach to the obtained regression results. Please revisit that and report your final results.

Author Response

- Please try to spell out any abbreviation at its 1st time eg DSM-IV and PTSD

Consistently with the reviewer’s remark, we have spelled all the abbreviations in the text at their first appearance (page 6, lines 233-234).

- In table 1: please delete the columns named "F (or chi-squared) and df". Please try to spell out any abbreviation in the table footnote.

We thank the Referee for the suggestion. Consistently, in Table 1 we have deleted the cited columns and have corrected the footnotes. (page 7-8).

Please try to report DASS-21 Total scores comparison of the 3 groups using boxplots and add overall p value inside the figure. Authors may do that with some other variables of their choice as well. Adding a graphical summary of their finding will be more attractive to the reader.

We agree with the reviewer’s cue and have included four boxplots (PANAS Positive Affect, PANAS Negative Affect, DASS-21 total score and IES-R total score) (Fig.1, page 10).

- Please expand the legend for table 3 and explain the results. It is better to report beta as a regression coeficient and report its 95%CI. Reader will not be interested the stepwise approach to the obtained regression results. Please revisit that and report your final results.

Accordingly with the Referee’s suggestion we have edited table 3 (page 11).

Reviewer 3 Report

The objectives of this study were to investigate the characteristics of post-covid brain fog and particularly its relationship with cognitive and neuropsychiatric symptoms and to establish the socio-demographic factors related to its occurrence. 

Thanks to the authors for their work which allows an objective diagnosis of these disorders. Interestingly, brain fog sensation does not necessarily coincide with Scc. Indeed it is one of the first studies to provide a particularly useful classification. 

The manuscript is well written and very clear. The tables are clear and easy to understand. 

The design of the study is not optimal but one can imagine the difficulty so I think the design is adapted to the question asked. 

All the information needed to understand the study is given in the introduction. The discussion is a bit long for my taste but allows a good interpretation of the results. 

Author Response

We thank the referee for the positive comments. We are glad that the aims of the study were clear and of interest. 

Consistently with the remark about the lenght of the Discussion, we have tried to be more coincise and have included a new paragraph to be more precise and effective in the outline of such a complex topic.

Reviewer 4 Report

Dear authors, thank you for the opportunity to get to know your work and congratulate you on it. This is, undoubtedly, a work of undeniable interest insofar as it not only systematises the concept of brain Fog and subjective cognitive complaints, but also establishes a relationship with neuropsychiatric and cognitive symptoms and with the socio-demographic variables.

However, there is a fundamental methodological issue that requires clarification. It has to do with the fact that the Covid-19 status of each participant is unknown. In fact, given the online nature of this study design, we could only appeal to what the patient would know about it. However, that is the condition of all other questions. That is, the researchers are limited to what the subjects answer without any possibility of further confirmation.

Under this condition it remains unclear whether brain fog and subjective cognitive complaints are a sequela of Covid-19 or a condition present in the population.

Although you address this issue in the chapter on limitations, namely in the third limitation, I think this is a structural aspect that deserves further explanation.

Author Response

Dear authors, thank you for the opportunity to get to know your work and congratulate you on it. This is, undoubtedly, a work of undeniable interest insofar as it not only systematises the concept of brain Fog and subjective cognitive complaints, but also establishes a relationship with neuropsychiatric and cognitive symptoms and with the socio-demographic variables.

However, there is a fundamental methodological issue that requires clarification. It has to do with the fact that the Covid-19 status of each participant is unknown. In fact, given the online nature of this study design, we could only appeal to what the patient would know about it. However, that is the condition of all other questions. That is, the researchers are limited to what the subjects answer without any possibility of further confirmation.

Under this condition it remains unclear whether brain fog and subjective cognitive complaints are a sequela of Covid-19 or a condition present in the population.

Although you address this issue in the chapter on limitations, namely in the third limitation, I think this is a structural aspect that deserves further explanation.

We thank the reviewer for the positive comments and we are glad that the aims of our study were of interest. Nonetheless, we agree with the reviewer that the issue of lack of information about previous participants’ COVID-19 infection is crucial and that it required a clearer explanation.

Thus, firstly we have better outlined the issue in the Methods section as follows: “We did not consider the subjects' actual previous infection due to COVID-19 as an inclusion criterion, given the web-based design of the research and the consequent inability to collect reliable information about this point. Many people who experienced mild or asymptomatic SARS-CoV-2 infections did not receive antigen testing at the time of acute infection [42]. Not to say, some long haulers might not have detectable antibodies to SARS-CoV-2 when the first serological test became available commercially [9]. All these considerations prevented an undoubtful attribution of later symptoms to SARS-CoV-2 infection. Nonetheless, some further remarks partially contain the relevance of this issue. Recent studies highlighted that post-COVID symptoms were not necessarily related to the severity of infection [43] and that similar long-term consequences were present in affected and not affected subjects [9,44,45]. Also, several studies on Long-COVID do not include a control group of uninfected subjects, thus preventing inferences about attributable risk [42].” (Page 4-5, lines 155-169).

Secondly, we have included in paragraph 4.2 (“The role of neuropsychiatric symptoms”) the following statement: “Thus, consistently with psychological models, we might speculate that FBF and PBF could develop without the infective chain of events. In our study, the rate of FBF and PBF cases amounts to 65%, and we had no information about previous COVID-19 infections in our sample. We cannot exclude a share of not infected subjects included in the FBF and PBF groups.”  (page 15, lines 577-581)

Finally, in the Limitations paragraph, we better outlined that “Fourth, as widely stressed above, in the Discussion section, we have no information about previous COVID-19 infections in our sample. Despite this, in our study, the post-pandemic outburst criterion relates FBF and PBF to pandemic events regarding infection and /or psychologically traumatic consequences. However, this speculation requires future accurate deepening. Further studies comparing a group of confirmed infected patients and a control group including uninfected subjects could better cast light on our results and deepen our conclusions on FBF and PBF”.  (page 17, lines 709-717).

Round 2

Reviewer 2 Report

Authors addressed my comments and it is my pleasure to accept their work.

Reviewer 4 Report

Dear authors, thank you very much for your answer and your changes to the paper, which undoubtedly improve and clarify it, making it more scientifically appealing and relevant. Congratulations for the excellent work.